# The Combination of Oolonghomobisflavan B and Diallyl Disulfide Induces Apoptotic Cell Death via 67-kDa Laminin Receptor/Cyclic Guanosine Monophosphate in Acute Myeloid Leukemia Cells

**Jaehoon Bae and Su-Jin Park *** 

Functional Biomaterial Research Center, Korea Research Institute of Bioscience and Biotechnology, 181 Ipsin-gil, Jeongeup-si 56212, Republic of Korea; baejae8@yahoo.co.jp
* Correspondence: sjpark@kribb.re.kr; Tel.: +82-(63)-570-5240

**Abstract:** Diallyl disulfide (DADS) is a well-known principal functional component derived from garlic (*Allium sativum*) that has various health benefits. Previously, we identified a 67-kDa laminin receptor, a receptor for oolong tea polyphenol oolonghomobisflavan B (OHBFB). However, its molecular mechanisms still remain to be elucidated. Here, we show that DADS synergistically enhanced the effect of the oolong tea polyphenol oolonghomobisflavan B (OHBFB), which induces apoptosis in acute myeloid leukemia (AML) cancer cells without affecting normal human peripheral blood mononuclear cells (PBMCs). The underlying mechanism of OHBFB-induced anti-AML effects involves the upregulation of the 67-kDa laminin receptor/endothelial nitric oxide synthase/cyclic guanosine monophosphate (cGMP)/protein kinase c delta (PKCδ)/acid sphingomyelinase (ASM)/cleaved caspase-3 signaling pathway. In conclusion, we show that the combination of OHBFB and DADS synergistically induced apoptotic cell death in AML cells through activation of 67LR/cGMP/PKCδ/ASM signaling pathway. Moreover, in this mechanism, we demonstrate DADS may reduce the enzyme activity of phosphodiesterase, which is a negative regulator of cGMP that potentiates OHBFB-induced AML apoptotic cell death without affecting normal PBMCs.

**Keywords:** oolonghomobisflavan B; diallyl disulfide; acute myeloid leukemia; cyclic guanosine monophosphate; acid sphingomyelinase

## 1. Introduction

Oolonghomobisflavan B (OHBFB) emerges as a distinctive component within oolong tea polyphenols renowned for its diverse health benefits [1]. Our prior research successfully identified the 67-kDa laminin receptor (67LR) as the specific cell surface receptor for OHBFB [2]. Importantly, 67LR exhibits overexpression in various cancer cell types [3], resulting in constrained plasma concentrations of polyphenols and, consequently, a limitation on their anti-cancer efficacy [4,5]. The molecular mechanisms initiated by OHBFB remain largely elusive, compelling us to delve into more effective therapeutic strategies for combating acute myeloid leukemia (AML). This study substantiates OHBFB's induction of anti-AML activity, thereby laying the groundwork for its clinical evaluation.

Cyclic guanosine monophosphate (cGMP) targets protein kinase G, while phosphodiesterase (PDE) acts as a negative regulator of cGMP [6–8]. The inhibition of PDE enzyme activity to activate the cGMP pathway has emerged as a novel therapeutic approach for various pathologies, including erectile dysfunction [9–12]. Recent studies have reported the use of innovative PDE5 inhibitors to treat a spectrum of pathologies.

Diallyl disulfide (DADS), an organosulfide derived from garlic (*Allium sativum*), serves as a principal functional component inhibiting PDE enzyme activity and inducing vasorelaxation [13,14]. Abundant studies provide evidence that DADS, a bioactive

garlic compound, offers a range of health benefits, notably including its anti-cancer potential [15–18]. Previously, we identified $H_2S$ as a gasotransmitter that potentiates cancer cell death in multiple myeloma cells [17]. Intracellular $H_2S$ suppresses phosphodiesterase enzyme activity, thereby elevating cyclic nucleotide levels and promoting cancer cell death [13,17]. Here, we showed that a garlic-derived bioactive garlic compound, DADS, induced intracellular $H_2S$ production in HL60 cells. Moreover, we administered a diet to mice that included DADS (2 mg/kg body weight) and tea extract (0.1% green tea extract). This combination, equivalent to the human intake of approximately only a cup of tea per day and two cloves of garlic, was given for 10 weeks. The tea polyphenols + DADS diet attenuated the HF/HS diet-induced upregulation of ALT and AST levels in mice, suggesting potential liver protection against damage induced by the HF/HS diet [18]. However, the anti-AML mechanism of DADS remains relatively unexplored. Our study assessed the impact of DADS on OHBFB-induced apoptotic cell death effects without affecting normal human peripheral blood mononuclear cells (PBMCs). Through isobologram analysis, we demonstrated that the combination of OHBFB and DADS indicated a synergistic induction of cell death effects in HL60 cells. Furthermore, our study highlights that DADS potentiates OHBFB-induced apoptotic cell death by activating the endothelial nitric oxide synthase/cGMP/protein kinase c delta/acid sphingomyelinase signaling pathway in AML cells while sparing normal PBMCs.

## 2. Materials and Methods

### 2.1. Materials

OHBFB was sourced from Nagara Science in Gifu, Japan, while DADS was procured from Cayman Chemical in Ann Arbor, MI, USA. Bay 41-2272 was supplied by Enzo Life Sciences in Crescent, Exeter, UK. Antibodies, including anti-PKC delta and anti-eNOS, were acquired from Abcam in Cambridge, MA, USA. The Phospho-PKC delta pSer664 antibody was obtained from Thermo Fisher Scientific in Waltham, MA, USA, and anti-Phospho-eNOS (pSer1177) antibody was purchased from BD Biosciences in San Jose, CA, USA. The anti-cleaved caspase-3 (Asp175) antibody was provided by Cell Signaling Technology in Danvers, MA, USA, while anti-β actin antibody was sourced from Sigma-Aldrich in Saint Louis, MO, USA. The anti-IgM antibody (A-7) was procured from Santa Cruz Biotechnology, and the anti-67LR antibody (MLuC5, anti-Laminin-R) was also obtained from Santa Cruz Biotechnology in Dallas, TX, USA. Primary PBMCs were generously provided by Takara in Shiga, Japan.

### 2.2. Cell Culture

AML cell line HL60 was maintained at $5 \times 10^4$ cells/well in IMDM supplemented with 1% FBS. Cultures were maintained at 37 °C with 100% humidity and 5% $CO_2$. The viability of HL60 cells was assessed after 72 h using the trypan blue exclusion analysis. Apoptotic cell death was evaluated through flow cytometric analysis. The cultured cells were co-stained with PI and Annexin V, and (early apoptotic) Anexin V+/PI− cells and (late apoptotic) Annexin V+/PI+ cells were identified using flow cytometry 72 h post-treatment with OHBFB and/or DADS. The analyses were performed using the VerseTM system from Becton Dickinson, located in Franklin Lakes, NJ, USA.

### 2.3. Western Blot Analysis

HL60 cells were plated at $5 \times 10^4$ cells/well or $1 \times 10^6$ cells/well and subjected to treatment with (5 μM) OHBFB and/or (10 μM) DADS for 3 h or 72 h, depending on the specific experiment. Post-treatment, cells were lysed in a lysis buffer (pH 7.5) comprising 50 mM Tris-HCl (pH 7.5), 30 mM $Na_4P_2O_7$, 150 mM NaCl, 50 mM NaF, 1% Triton X-100, 2 mg/mL aprotinin, 1 mM phenylmethanesulfonyl fluoride, 1 mM ethylenediaminetetraacetic acid, and 1 mM pervanadate. Subsequently, SDS-PAGE was conducted following established protocols and sourced from Sigma-Aldrich in Saint Louis, MO, USA [19].

### 2.4. Quantitative Reverse Transcription PCR (qRT-PCR)

HL60 cells were incubated in the absence or presence of 5 μM OHBFB and/or 10 μM DADS. After a 3 h exposure, cells underwent three successive washes with phosphate-buffered saline. Subsequently, cells were harvested by centrifugation at 500× *g* for 5 min, and total RNA extraction followed the manufacturer's protocol employing the Qiagen RNeasy Mini Kit (QIAGEN, Hilden, Germany). Primers designed for homo sapiens ribosomal protein SA (67LR) were as follows: 5′-GCAGCAGGAACCCACTTAGG-3′ (forward) and 5′-GCAGCAGCAAACTTCAGCAC-3′ (reverse); primer human GAPDH: 5′-CCACTCCTCCACCTTTGACG-3′ (forward) and 5′-CCACCACCCTGTTGCTGTAG-3′ (reverse). cDNA synthesis from total RNA was accomplished using the cDNA Master Mix (Applied Biosystems, Foster City, CA, USA). qRT-PCR was performed using 2 μL cDNA and SYBR Green PCR 2 × Master Mix (40 cycles of (15 s) at 95 °C, (60 s) at 60 °C) from Applied Biosystems (Foster City, CA, USA). Data analysis utilized (v2.1) StepOne software from Applied Biosystems (Foster City, CA, USA).

### 2.5. cGMP Assays

cGMP production levels were analyzed using the AlphaScreen cGMP kit from PerkinElmer (Waltham, MA, USA). HL60 cells were treated with OHBFB and/or DADS for an hour. Plates were evaluated using the EnVisionTM Plate Reader (PerkinElmer).

### 2.6. Acid Sphingomyelinase (ASM) Activity Measurement

HL60 cells were lysed by incubating in a lysis buffer (4 °C) for one hour. Following incubation, the lysate underwent centrifugation (20 min) at 12,000× *g*. The resulting supernatant was then mixed with a substrate buffer containing BODIPY-C12 (400 pmol) sphingomyelin in dH2O (1% Triton X-100 and 200 mM sodium acetate) and incubated for 24 h at 37 °C.

### 2.7. Statistical Analysis

The results are presented as mean ± standard error of the mean (SEM). Isobologram analysis and determination of 50% inhibitory concentrations (IC50) were conducted using 2.0 Calcusyn software (Biosoft), Cambridge, UK. Significance of differences was evaluated using Tukey's test, and all statistical analyses were carried out with KyPlot 6.0, version 6.0.2 software provided by KyensLab Inc., Tokyo, Japan.

## 3. Results

### 3.1. Synergistic Induction of Cell Death in HL60 Cells by the Combination of OHBFB and DADS

OHBFB induces an anti-cancer effect by targeting the 67LR [2]. However, polyphenols exhibit limited activity at physiological concentrations [4]. We evaluated the anti-AML effect of OHBFB and/or DADS in HL60 cells. We found that the IC50 of OHBFB alone was 10.8 μM, and the IC50 of DADS alone was 55.8 μM in HL60 cells (Figure 1A,B). Interestingly, the IC50 of OHBFB was 5.7 μM when treated with DADS at 10 μM (Figure 1C). Moreover, when treated with 20 μM DADS, the IC50 of OHBFB further decreased to 3.1 μM (Figure 1D). These results indicate that DADS potentiated the OHBFB-induced cell death effect in HL60 cells. The isobologram analysis curve demonstrated synergistic effects in HL60 cells with the combination of OHBFB and DADS (Figure 1E). Moreover, OHBFB and/or DADS induced a cell death effect in HL60 cells while exhibiting no impact on normal peripheral blood mononuclear cells (PBMCs) (Figure 1F). These findings suggest that the combination of OHBFB and DADS synergistically induces cancer-selective cell death without affecting normal PBMCs.

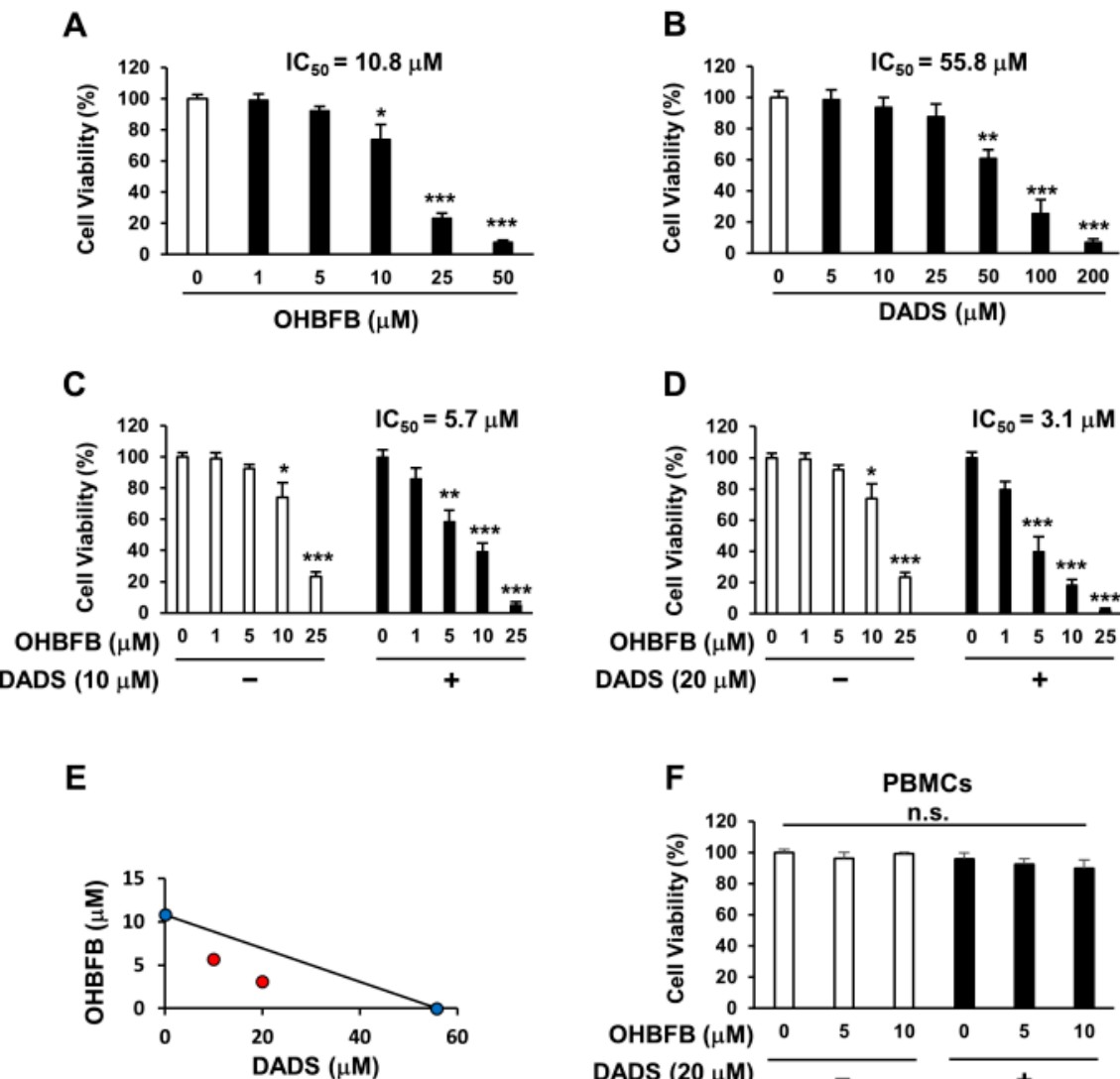

**Figure 1.** Synergistic induction of cell death in HL60 cells by the combination of OHBFB and DADS. (**A–D**) HL60 cells underwent treatment with OHBFB and/or DADS at the specified concentrations for 72 h, and subsequent measurements were taken to assess the cell viability of HL60 cells. (**E**) The synergistic impact of OHBFB and DADS was gauged through isobologram analysis. (Red circle: combination treatment of OHBFB and DADS. Blue circle: OHBFB alone or DADS alone treatment) (**F**) DADS (20 μM) and/or OHBFB (5 μM) were administered to normal human peripheral blood mononuclear cells (PBMCs) for 72 h, and the cell viability of normal PBMCs was then evaluated. n.s.: not significant. The data are presented as mean ± the standard error of the mean (SEM) ($n = 3$). * $p < 0.05$; ** $p < 0.01$; *** $p < 0.001$.

### 3.2. Combination of OHBFB and DADS Induces Apoptotic Cell Death

To explore the potential apoptosis induction in HL60 cells by the OHBFB and DADS combination, HL60 cells were subjected to 5 μM OHBFB and/or 10 μM DADS. The concurrent application of 5 μM OHBFB and 10 μM DADS initiated apoptosis; conversely, the individual treatments with either 5 μM OHBFB or 10 μM DADS alone did not induce apoptosis (Figure 2A). Furthermore, the combined exposure to OHBFB and DADS resulted in caspase-3 cleavage (Figure 2B).

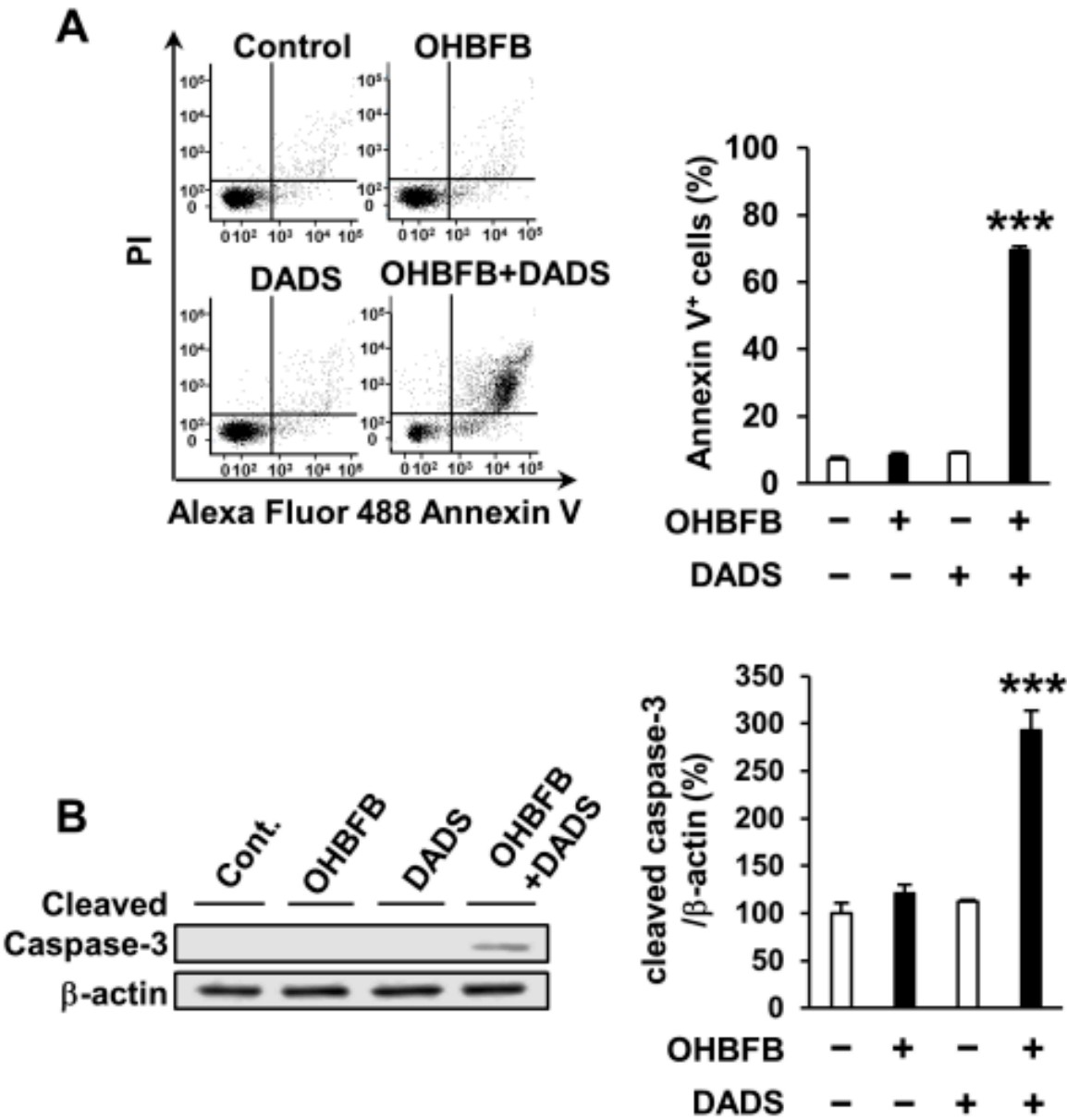

**Figure 2.** Combination of oolonghomobisflavan B (OHBFB) and diallyl disulfide (DADS) induces apoptotic cell death. (**A**) HL60 cells were treated with OHBFB (5 μM) and/or DADS (10 μM) for 72 h. Apoptotic cells were stained with propidium iodide and Annexin V–Alexa Fluor 488. (**B**) Cleaved caspase-3/β-actin levels were measured. Data are presented as mean ± SEM (*n* = 3). *** *p* < 0.001.

### 3.3. Vardenafil Synergistically Potentiates Cell Death Induced by OHBFB

Vardenafil (VDN) is a selective PDE5 inhibitor. To investigate whether the PDE5 inhibition can potentiate OHBFB-induced cell death in HL60 cells, the cells were treated with OHBFB and/or VDN. The IC50 of VDN alone was found to be 27.5 μM (Figure 3A), while the IC50 of OHBFB alone was 10.8 μM (Figure 1A) in HL60 cells. Interestingly, the IC50 of OHBFB reduced to 7.2 μM when treated with VDN at 1 μM (Figure 3B). Moreover, with a 5 μM VDN treatment, the IC50 value of OHBFB further decreased to 3.9 μM (Figure 3C). The isobologram analysis curve demonstrated synergy in HL60 cells with the combination of OHBFB and the VDN, a PDE5 inhibitor (Figure 3D). These results indicate that the PDE5 inhibition potentiates the OHBFB-induced cell death effect in HL60 cells.

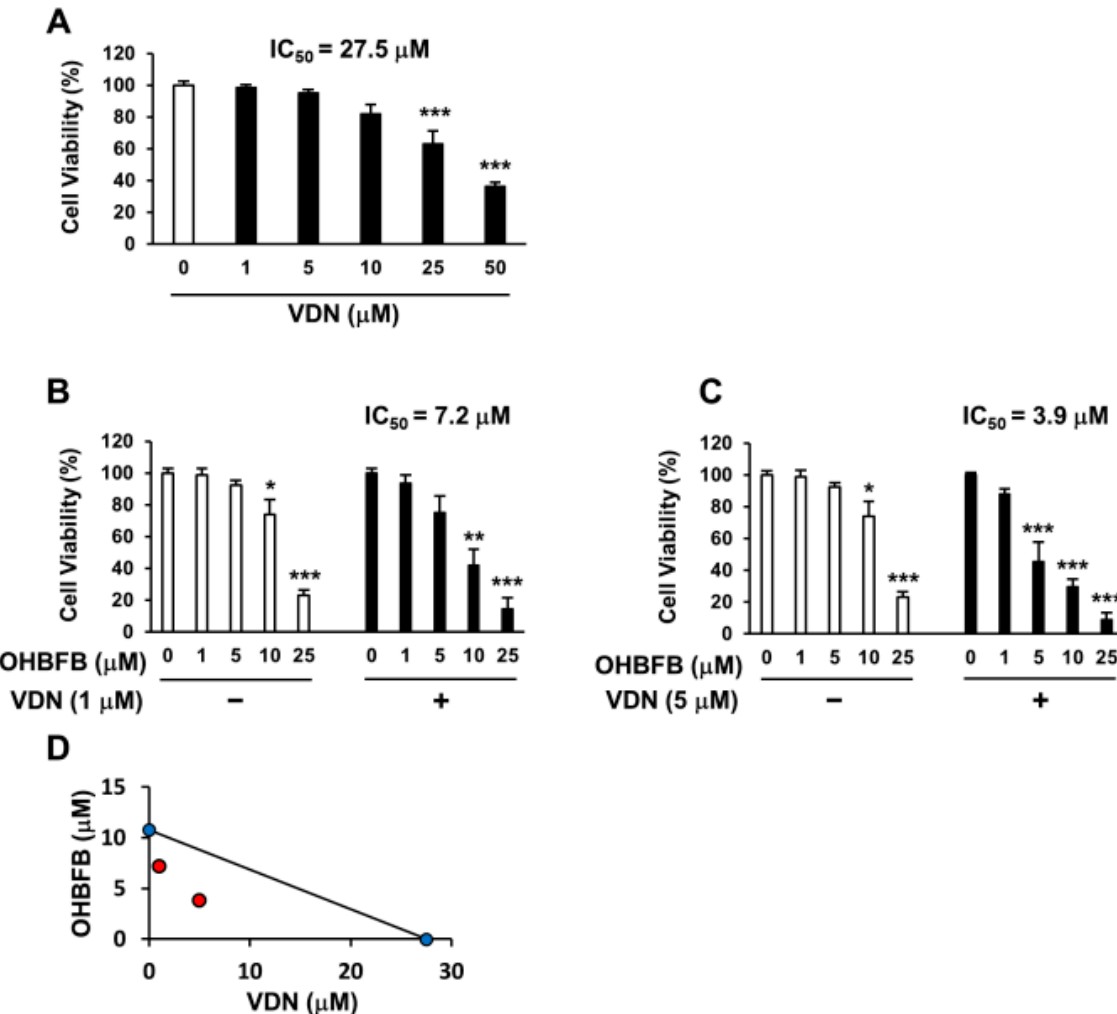

**Figure 3.** Vardenafil (VDN) synergistically potentiates cell death caused by oolonghomobisflavan B (OHBFB). (**A**–**C**) HL60 cells were cultured with VDN and/or OHBFB for 72 h at the indicated concentrations. After 72 h, cell viability was measured. (**D**) The synergistic effect of VDN and OHBFB was measured. Red circle: combination treatment of OHBFB and DADS. Blue circle: OHBFB alone or DADS alone treatment. Data are presented as mean ± SEM (*n* = 3). * *p* < 0.05; ** *p* < 0.01; *** *p* < 0.001.

### 3.4. DADS Potentiates OHBFB-Induced Activation of cGMP Pathway

To assess the impact of DADS on the upstream and downstream signals of cGMP, we examined its effects on OHBFB-derived eNOS phosphorylation, PKCδ phosphorylation, cGMP production, and ASM activity. While OHBFB induces eNOS phosphorylation at Ser1177, DADS did not influence the OHBFB-induced eNOS phosphorylation (Figure 4A). PDE5, an enzyme that inhibits intracellular cGMP production [10,11], was also investigated. We observed that 10 μM OHBFB increased intracellular cGMP levels, whereas 0–5 μM OHBFB had no significant effect on HL60 cells (Figure 4B). Additionally, our data revealed that DADS potentiated OHBFB-induced cGMP levels (Figure 4C). These findings suggest that DADS reduces cGMP-PDE enzyme activity. Furthermore, the combination of OHBFB and DADS upregulated PKCδ phosphorylation (Figure 4D). We also observed that the combination of OHBFB and DADS induced ASM activity (Figure 4E). Therefore, we explored the potential of DADS and BAY 41-2272 in inducing cGMP-dependent signaling and ASM activity in HL60 cells. Bay 41-2272 is a soluble guanylate cyclase agonist, also recognized as an activator of intracellular cGMP production [3]. The combined application of DADS and BAY 41-2272 demonstrated increased ASM activity in HL60 cells (Figure 4F). These

findings suggest that DADS enhances OHBFB-induced apoptotic cell death through the involvement of eNOS/cGMP/PKCδ/ASM signaling in AML cells.

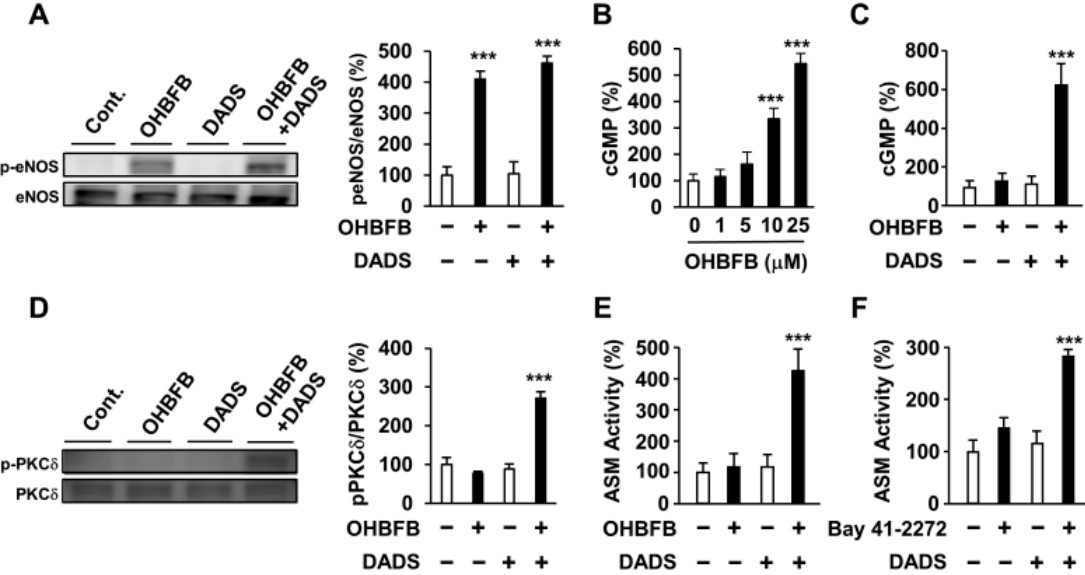

**Figure 4.** Diallyl disulfide (DADS) potentiated oolonghomobisflavan B (OHBFB)-induced activation of the cyclic guanosine monophosphate (cGMP) pathway. (**A**) HL60 cells were treated with OHBFB (5 μM) and/or DADS (10 μM). After one hour, phosphorylation eNOS at Ser1177 was measured. (**B**) HL60 cells were treated with OHBFB. After 1 h, intracellular cGMP levels were determined. (**C**) HL60 cells were treated with OHBFB (5 μM) and/or DADS (10 μM). After one hour, cGMP levels were determined. (**D**) HL60 cells were treated with OHBFB (5 μM) and/or DADS (10 μM). After 3 h, PKCδ phosphorylation at Ser664 was measured. (**E**) HL60 cells were treated with OHBFB (5 μM) and/or DADS (10 μM). After 72 h, ASM activity was assessed using TLC. (**F**) HL60 cells were treated with the cGMP activator Bay 41-2272 (1 μM) and/or DADS (10 μM). After 72 h, ASM activity was measured. Data are presented as mean ± SEM ($n = 3$). *** $p < 0.001$.

### 3.5. Combined OHBFB and DADS Induce Cell Death via 67LR in HL60 Cells without Affecting Normal Cells

To elucidate the involvement of 67LR in the synergistic anti-cancer effect of the OHBFB and DADS combination in HL60 cells, the cells underwent treatment with either a control IgM antibody or an anti-67LR antibody (MLuC5). The use of a control IgM antibody or anti-67LR antibody (MLuC5) is a well-established approach for assessing the contribution of 67LR to anti-cancer effects [3]. Our experimental results demonstrated that pre-treatment with the anti-67LR antibody mitigated the OHBFB- and DADS-induced cell death effect in HL60 cells (Figure 5A). Furthermore, we investigated the impact of OHBFB and DADS on the expression levels of 67LR. Notably, our findings indicated that neither OHBFB nor DADS, alone or in combination, influenced the mRNA expression levels of 67LR in HL60 cells (Figure 5B). A schematic representation was provided, illustrating how DADS enhances OHBFB-induced apoptotic cell death in HL60 cells (Figure 5C). Taken together, these results suggest that the combined action of OHBFB and DADS induces an anti-cancer effect through 67LR without affecting the mRNA expression levels of 67LR.

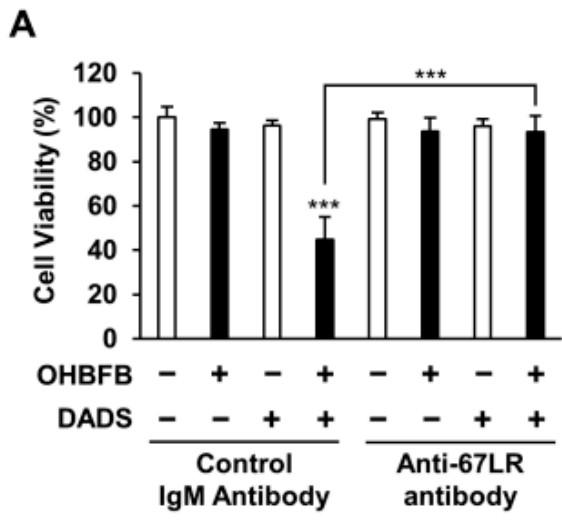

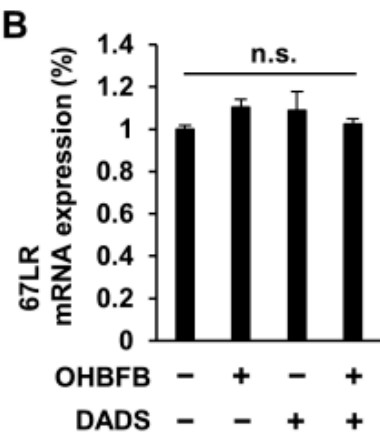

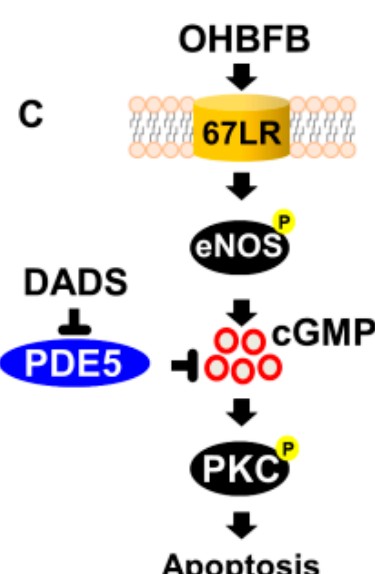

**Figure 5.** Combined OHBFB and DADS induce cell death via 67LR in HL60 cells without affecting normal cells. (**A**) After a 2 h pre-treatment with control IgM antibody or MLuC5 (anti-67LR antibody), HL60 cells were exposed to 5 μM OHBFB and/or 10 μM DADS for 72 h. Cell viability was evaluated using the trypan blue method (*n* = 3). (**B**) Primary peripheral blood mononuclear cells (PBMC) were subjected to treatment with 5 μM OHBFB and/or 10 μM DADS for 72 h, and the mRNA expression of 67LR was quantified using real-time PCR analysis (*n* = 3). n.s.: not significant. (**C**) The signaling pathways of OHBFB and DADS in HL60 cells are depicted in the schematic diagram. Data are presented as mean ± SEM (*n* = 3). *** $p < 0.001$.

## 4. Discussion

Oolong tea has been demonstrated to offer various health benefits [1]. OHBFB, a polyphenolic compound found in oolong tea, is a dimer of epigallocatechin 3-O-gallate (EGCG), the most potent component in green tea. Previously, we have identified 67LR, a cell surface protein that serves as a receptor for OHBFB [2]. 67LR is overexpressed in several types of cancer cells [3]. Consequently, not only are the plasma concentrations of polyphenols limited, but the molecular mechanism remains poorly understood. In this study, we demonstrated that the combination of 5 μM OHBFB and 10 μM DADS induces

apoptotic cell death through the activation of the cGMP signaling pathway. This provides a foundation for further evaluating its clinical potential.

Diallyl disulfide (DADS), a bioactive garlic compound recognized for its anti-cancer potential, is a primary organosulfur component derived from garlic (*Allium sativum*) and functions as a hydrogen sulfide ($H_2S$) donor. Intracellular $H_2S$ inhibits PDE activity and enhances cyclic nucleotide levels, leading to vasorelaxation [13,14]. Numerous studies have substantiated that DADS elicits a range of health-promoting effects, notably including its anti-cancer properties [14–17]. However, the mechanisms underlying the anti-acute myeloid leukemia (AML) effects of DADS remain poorly understood. Our study aimed to evaluate the impact of DADS on the anti-AML effects of OHBFB. DADS, a prominent garlic-derived organosulfide, functions as a $H_2S$ donor [13,20]. $H_2S$ serves as a regulator in cardiovascular, nervous, and immune systems [21–24]. The therapeutic potential of numerous $H_2S$ donors has been reported [25,26]. $H_2S$ may be derived from DADS through a glutathione-dependent mechanism in human red blood cells or in rat aorta [20].

Furthermore, $H_2S$ serves as a PDE5 inhibitor, specifically targeting the negative regulation of cGMP [14]. Clinical studies have demonstrated the effectiveness of PDE5 inhibitors in treating various diseases, including erectile dysfunction, heart failure, and pulmonary hypertension [27–29]. Additionally, the activation of the cGMP signaling pathway plays a pivotal role in anti-cancer activities [3]. However, the impact of polyphenols is limited by plasma concentration [4,5], and the mechanism underlying the anti-AML effects of DADS and OHBFB remains poorly understood. We observed that individual treatments with either OHBFB or DADS did not influence cGMP levels. However, treatment with a combination of 5 μM OHBFB and 10 μM DADS resulted in an upregulation of intracellular cGMP levels. These findings suggest that DADS potentiates OHBFB-induced AML cell death, possibly by inhibiting cGMP-PDE enzyme activity.

The gasotransmitter nitric oxide (NO) is produced by eNOS, and it regulates soluble guanylyl cyclase (sGC), thereby inducing intracellular cGMP production [30–32]. We confirmed that OHBFB induced eNOS phosphorylation at Ser1177; however, treatment with DADS alone did not appear to have the same effect. These results suggest that DADS potentiates AML cell death induced by OHBFB by enhancing downstream cGMP signaling but not upstream signaling.

Activation of acid sphingomyelinase (ASM) is implicated in cancer cell apoptosis induced by ultraviolet radiation, cisplatin, and oxidative stress [33,34]. ASM serves as an indispensable mediator of cGMP-initiated apoptosis in cancer cells [19,35]. However, the molecular mechanisms involved in cGMP-induced cell death in AML are not fully clear. Here, we demonstrated that intracellular cGMP production is sufficient to induce ASM activity in HL60 cells. Our results suggest that ASM activity partially contributes to these effects in HL60 cells. Moreover, DADS potentiates AML apoptotic cell death induced by OHBFB at only 5 μM, which is closest to the physiological concentration, through activation of the cGMP/ASM/cleaved caspase-3 signaling pathway.

We have demonstrated the intracellular generation of $H_2S$ by DADS in HL60 cells (Supplemental Figure S1). Additionally, our confirmation revealed that the $H_2S$ donor, DADS, enhances ASM activation prompted by the NO-independent sGC activator BAY 41-2272. Notably, the OHBFB-induced eNOS/cGMP/ASM/caspase-3 axis emerges as a crucial player in driving apoptotic cell death in HL60 cells. Significantly, a solitary exposure to DADS does not seem to initiate the phosphorylation of eNOS at Ser1177 in vitro. Additionally, we confirmed that DADS does not influence the OHBFB-induced phosphorylation of eNOS at Ser1177. These results imply that the $H_2S$ donor intensifies the apoptotic cell death effect of OHBFB by enhancing downstream cGMP, all the while leaving eNOS phosphorylation at Ser1177 unaffected in HL60 cells without impacting normal human peripheral blood mononuclear cells (PBMCs) under the concentrations tested (Supplemental Figure S2). Additionally, $H_2S$ is acknowledged as an inhibitor of PDE5 activity [31]. Crucially, the presence of OHBFB did not impact the production of $H_2S$ in HL60 cells. Consequently, these findings suggest that the intracellular $H_2S$ generated by

DADS intensifies the apoptotic cell death effect induced by OHBFB through the inhibition of cGMP-PDE enzyme activity in HL60 cells.

In summary, our findings suggest that DADS amplifies the apoptotic cell death effect induced by OHBFB, concomitant with the activation of the eNOS/cGMP/PKCδ/ASM signaling pathway in HL60 cells. Given that the effective experimental concentrations of the combination treatment with these compounds are lower than when treated alone, these results merit particular consideration. Moreover, the combination of OHBFB and DADS may be considered a promising candidate for the development of natural therapeutic drugs against AML. Further studies are needed to confirm therapeutic efficacy, long-term effects, and the potential for resistance development for safety (in vivo).

**Supplementary Materials:** The following supporting information can be downloaded at: https://www.mdpi.com/article/10.3390/cimb46030154/s1, Figure S1: The effect of OHBFB or DADS on intracellular H2S production. HL60 cells are treated with OHBFB or DADS for 3 hours at the indicated concentrations, and intracellular H2S are measured by highly selective fluorescence probe HSip-1 (lex 491 nm, lem 516 nm). Data are presented as mean $\pm$ SEM ($n$ = 4). *** $p < 0.001$; Figure S2: Normal human peripheral blood mononuclear cells (PBMCs) are treated with DADS and/or OHBFB for 72 h at indicated concentrations, and the cell viability of Normal PBMCs was measured. Data are presented as mean $\pm$ the standard error of the mean (SEM) ($n$ = 3).

**Author Contributions:** J.B. and S.-J.P. contributed to the research idea and design. J.B. and S.-J.P. created the search strategy. J.B. and S.-J.P. screened titles, abstracts, and full texts. J.B. and S.-J.P. contributed to data extraction and quality assessment. J.B. and S.-J.P. contributed to the statistical analysis and interpretation of data. J.B. wrote the first draft of the manuscript. J.B. and S.-J.P. edited the draft of the manuscript. All authors have read and agreed to the published version of the manuscript.

**Funding:** This work was supported by the National Research Foundation of Korea (NRF) grant funded by the Korean government (MSIT) (NRF-2021R1C1C2095006) and the KRIBB Research Initiative Program (KGM5242423), Republic of Korea.

**Institutional Review Board Statement:** Not applicable.

**Informed Consent Statement:** Not applicable.

**Data Availability Statement:** The data that support the findings of this study are available upon request from the corresponding author.

**Conflicts of Interest:** The authors declare no conflicts of interest.

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
