# Peer review of "The Combination of Oolonghomobisflavan B and Diallyl Disulfide Induces Apoptotic Cell Death via 67-kDa Laminin Receptor/Cyclic Guanosine Monophosphate in Acute Myeloid Leukemia Cells"

_cimb, doi:10.3390/cimb46030154_

Round 1
Reviewer 1 Report
Comments and Suggestions for Authors
The authors in the present study have shown that in acute myeloid leukemia cells, diallyl disulfide (DADs) and oolonghomobisflavan B (OHBFB) together cause apoptotic cell death via the 67-Kda laminin receptor and cyclic guanosine monophosphate. They showed that the combination of this drug do not have any deleterious effect on normal cells (PBMCs) I have following concern
1. Please make a separate section for materials as it looks like it is merged in cell culture section.
2. It would be interesting if the author would elaborate the beneficial role of DADs in context of different diseases. The Introduction is too short, it needs more description on the background of the study
3. The authors were describing the synergistic effect DADS and OHBFB and they suddenly switched to Vardenafil in section 3.3 and Figure 3. Nowhere in text in results the role of Vardenafil in the present study context is confusing. The whole figure 3 results are missing.
4. There is a lot of ithenticate similarity is over 40 percent. Which is a lot! The authors should sincerely consider working on this.
Comments on the Quality of English Language
English language needs editing. Its not clear at several places
Author Response
# Reviewer 1
The authors in the present study have shown that in acute myeloid leukemia cells, diallyl disulfide (DADs) and oolonghomobisflavan B (OHBFB) together cause apoptotic cell death via the 67-Kda laminin receptor and cyclic guanosine monophosphate. They showed that the combination of this drug do not have any deleterious effect on normal cells (PBMCs) I have following concern
- Please make a separate section for materials as it looks like it is merged in cell culture section.
Response 1: In accordance with the Reviewer’s comment, we made a separate section for materials.
- It would be interesting if the author would elaborate the beneficial role of DADs in context of different diseases. The Introduction is too short, it needs more description on the background of the study
Response 2: Thank you for your helpful suggestions. In accordance with the reviewer's comments, we have added more description on the background of the study in Introduction section as described below.
“In a previous study, we administered a diet to mice that included DADS (2 mg/kg body weight) and tea extract (0.1% green tea extract). This combination, equivalent to the human intake of approximately only a cup of tea per day and 2 cloves of garlic, was given for 10 weeks. The tea polyphenols + DADS diet attenuated the HF/HS diet-induced upregulation of ALT and AST levels in mice. These findings also suggest that this combination may protect the liver from damage induced by the HF/HS diet. [17].”
“We demonstrated, through isobologram analysis, that the combination of OHBFB and DADS indicated a synergistic induction of cell death effects in HL60 cells.”
- The authors were describing the synergistic effect DADS and OHBFB and they suddenly switched to Vardenafil in section 3.3 and Figure 3. Nowhere in text in results the role of Vardenafil in the present study context is confusing. The whole figure 3 results are missing.
Response 3: In accordance with the Reviewer’s comment, we have added the role of Vardenafil and figure 3 results in section 3.3 as described below.
“Vardenafil (VDN) is a selective PDE5 inhibitor. To investigate whether the inhibition of PDE5 can potentiates OHBFB-induced apoptotic cell death in HL60 cells, the cells were treated with OHBFB and/or VDN. We found that the IC50 of VDN alone was 27.5 uM (Figure 3A) and IC50 of OHBFB alone was 10.8 uM (Figure 1A) in HL60 cells. Interestingly, the IC50 of OHBFB was 7.2 uM when treated with VDN at 1 uM (Figure 3B). Moreover, when treated with 5 uM VDN, the IC50 value of OHBFB are further decreased to 3.9 uM (Figure 3C). Isobologram analysis curve showed that the combination of OHBFB and PDE5 inhititor VDN was synergetic in HL60 cells (Figure 3D). These results indicate that inhibition of PDE5 potentiated the OHBFB-induced cell death effect in HL60 cells.”
- There is a lot of ithenticate similarity is over 40 percent. Which is a lot! The authors should sincerely consider working on this.
Response 4: Thank you for your comment. Accordingly, we amended our manuscript and checked ithenticate similarity.

Reviewer 2 Report
Comments and Suggestions for Authors
· The paper does not provide detailed information about the molecular mechanisms of OHBFB-induced anti-AML effects and the specific role of DADS in this mechanism.
· The study does not explore the potential side effects or toxicity of the combination of OHBFB and DADS.
· The paper does not discuss the optimal dosage or concentration of OHBFB and DADS for inducing apoptotic cell death in AML cells.
· The study does not investigate the long-term effects or the potential for resistance development to the combination treatment.
· Cell line specificity may not fully represent diverse cancer types.
· The concentration-dependent effects and potential toxicities not fully explored.
· Limited assessment of off-target effects on normal cell populations.
· Absence of in vivo studies to confirm therapeutic efficacy and safety.
Comments on the Quality of English LanguageNeed extensive revisions
Author Response
# Reviewer 2
- The paper does not provide detailed information about the molecular mechanisms of OHBFB-induced anti-AML effects and the specific role of DADS in this mechanism.
Response 1: Thank you for your comment. In accordance with the Reviewer’s comment, we have added more description on the background of the study in Discussion section as described below.
“We show DADS produced intracellular H2S in HL60 cells (Supplemental Figure S1). Furthermore, we confirmed that the H2S donor DADS potentiated ASM activa-tion induced by a NO-independent sGC activator BAY 41-2272. The OHBFB-induced eNOS/cGMP/ASM/caspase-3 axis plays an important role in apoptotic cell death in HL60 cells. Single DADS treatment does not appear to induce phosphorylation of eNOS at Ser1177 in vitro. We also confirmed that DADS did not have an impact on OHBFB-elicited phosphorylation of eNOS at Ser1177. These results suggest that H2S donor potentiates the apoptotic cell death effect of OHBFB by enhancing the down-stream of cGMP but not affecting eNOS phosphorylation at Ser1177 in HL60 cells. Moreover, H2S is known as an inhibitor of PDE5 activity31. Furthermore, OHBFB did not affect H2S production in HL60 cells. These results suggest that intracellular H2S produced by DADS potentiates OHBFB-induced apoptotic cell death effect via inhi-bition of cGMP-PDE enzyme activity in HL60 cells.”
- The study does not explore the potential side effects or toxicity of the combination of OHBFB and DADS.
Response 2: Thank you for your helpful suggestions. Here we used natural compound oolong tea-derived compound OHBFB (5uM) or/and garlic-derived compound DADS (10uM) in vitro.
Previously study, we fed the mice a DADS-containing diet (2 mg/kg body weight) and tea extract (0.1 % green tea extract). This combination, equivalent to the human intake of approximately only 1 cup of green tea per day and 2 cloves of garlic.
The tea polyphenols + DADS diet (for 10 weeks) attenuated HF/HS diet -induced upregulation of ALT and AST levels in mice. These findings also suggested that this combination may be protect the liver from HF/HS diet-induced damage. Hepatic steatosis is also triggered by hyperinsulinemia, which leads to the induction of cytotoxic factors including inflammatory cytokines, lipid peroxidation, and causes transaminitis. The tea polyphenols +DADS diet also had an effect of lowering the serum cholesterol level and downregulating the cholesterol synthesis-related genes (HMGCR, HMGCS) in the liver of HF/HS diet-fed mice. Because HMGCR is the rate-controlling enzyme of the mevalonate pathway that is involved in cholesterol synthesis, the tea polyphenols +DADS diet may normalize hepatic lipid metabolism and TC levels by regulating HMGCR expression in the liver, followed by suppression of fat accumulation.
In accordance to the Reviewer’s comment, we have added in introduction section as described below.
“In a previous study, we administered a diet to mice that included DADS (2 mg/kg body weight) and tea extract (0.1% green tea extract). This combination, equivalent to the human intake of approximately only a cup of tea per day and 2 cloves of garlic, was given for 10 weeks. The tea polyphenols + DADS diet attenuated the HF/HS diet-induced upregulation of ALT and AST levels in mice. These findings also suggest that this combination may protect the liver from damage induced by the HF/HS diet.”
Also, as your comment, we have added in discussion section as described below.
“Further studies are needed to confirm therapeutic efficacy, long-term effect and potential for resistance development for safety (in vivo).”
- The paper does not discuss the optimal dosage or concentration of OHBFB and DADS for inducing apoptotic cell death in AML cells.
Response 3: Thank you for your helpful suggestions. In accordance to the reviewer's comments, we have added more description on the result section as described below.
To investigate whether the DADS can potentiates OHBFB-induced apoptotic cell death in HL60 cells, the cells were treated with OHBFB and/or DADS.
“We found that the IC50 of OHBFB alone was 10.8 mM, and the IC50 of DADS alone was 55.8 mM in HL60 cells (Figure 1A, B). Interestingly, the IC50 of OHBFB was 5.7 mM when treated with DADS at 10 mM (Figure 1C). Moreover, when treated with 20 mM DADS, the IC50 of OHBFB further decreased to 3.1 mM (Figure 1D). These results show that DADS potentiated the OHBFB-induced cell death effect in HL60 cells.”
Moreover, our result showed Isobologram analysis curve indicated that the combination of OHBFB and DADS was synergetic in HL60 cells (Figure 1E).
- The study does not investigate the long-term effects or the potential for resistance development to the combination treatment.
Response 4: Thank you for your comment. In accordance with the Reviewer’s comment, we have added more description in discussion section as described below.
“Further studies are needed to confirm therapeutic efficacy, long-term effect and potential for resistance development for safety (in vivo).”
- Cell line specificity may not fully represent diverse cancer types.
Response 5: As your comment, we amended manuscript as described below.
“anti-AML effect” to “cell death effect in HL60 cells” or “apoptotic cell death in HL60 cells.”
- The concentration-dependent effects and potential toxicities not fully explored.
Response 6: Thank you for your comment. In accordance with the Reviewer’s comment, we amended our manuscript, and described the concentration-dependent effects of intracellular cGMP production (added the experiments for the 25 mM OHBFB treatment) at Figure 4B and supplemental figure 2 for the effects on normal cells
- Limited assessment of off-target effects on normal cell populations.
Response 7: Thank you for your comment. Accordingly, we amended supplemental figure 2 as described below.
Normal human peripheral blood mononuclear cells (PBMCs) are treated with DADS and/or OHBFB for 72 h at concentration-dependent effect, and the cell viability of Normal PBMCs was measured.
We showed that OHBFB and/or DADS induced cell death effect in HL60 cells without cell death effect to normal peripheral blood mononuclear cells (PBMCs)
- Absence of in vivo studies to confirm therapeutic efficacy and safety.
Response 8: Thank you for your comment. We amended our manuscript in discussion section as described below.
Our study demonstrated that “DADS potentiates OHBFB-induced apoptotic cell death effect accompanied by the activation of the eNOS/cGMP/PKCδ/ASM signaling pathway in HL60 cells. As the effective experimental concentrations of combination treatment these compounds are low concentrations than treat alone, the results are worthy of particular consideration. Moreover, the combination of OHBFB and DADS may be considered as promising candidates for the development of natural therapeutic drugs against AML. Further studies are needed to confirm therapeutic efficacy, long-term effect and potential for resistance development for safety (in vivo).”

Reviewer 3 Report
Comments and Suggestions for Authors
There is a need for attention to the statistical aspect of the study.
Check carefully for example SEM. This stands for standard error of the mean.
Pay attention to details.
Comments on the Quality of English Language
None. Read carefully.
Author Response
# Reviewer 3
- There is a need for attention to the statistical aspect of the study. Check carefully for example SEM. This stands for standard error of the mean.
Response 1: Thank you for your comment. Accordingly, we checked for SEM as described below in figure caption 1.
“the standard error of the mean (SEM).”
Reviewer 4 Report
Comments and Suggestions for Authors
The investigated phenomenon, namely the synergistic effect of two natural substance on apoptosis has of moderate interest, especially because it was investigated only on one cell type. However the following of the signalization route experimentally is valuable.
The experimental set-up is clear and the presentation of the results is good. There is one point where the article could be improved - if the effect of DADS on cGMP levels could be demonstrated, if DADS is an inhibitor of PDE5 then this effect should be shown.
The English of the article is problematic, with many sentences difficult to understand, confusing structure of sentences.
The logic of the introduction is somewhat lacking. In line 33, the claim that OHBFB would have anti-AML effects is not convincing based on only in vitro results. In the section after line 36, the correlation between the 67LR receptor and cGMP levels is missing. It is not clear why erectile dysfunction is highlighted in relation to PDE5 activity in a cancer related article.
In the methodology section, the description of the statistical parameters is missing, how many independent experiments were performed in each case.
For the results section:
The term physiological concentration of OHBFB is not meaningful. Line 118 "DADS potentiated AML cell death caused by OHBFB, at an IC50 of 5.7 µM or 3.1 µM" is not understandable, presumably DADS concentrations are missing.
Slightly different parameters were used in the PBMCS cell viability analysis. A related question is what is the toxicity when using 25uM OHBFB? What is the reason for looking at viability at 96h for HL-60 cells and 72h for PBMCS cells?
In figure caption, the SEM abbreviation is probably standard error of mean correctly
cGMP-PDE enzyme activity - not meaningful term
line 170 BAY 41-2272 - cell biological activity of this substance should be added in the text somewhere
The discussion refers to many things, could be a bit more concise. The exaggerated wording of the last sentences overstates the value of simple in vitro experiments. Nor is "physiological concentration" meaningful here.
Comments on the Quality of English Language
The investigated phenomenon, namely the synergistic effect of two natural substance on apoptosis has of moderate interest, especially because it was investigated only on one cell type. However the following of the signalization route experimentally is valuable. In the literature both of the compounds were investigated alone in this respect, the novelty is in the combination.
The experimental set-up is clear and the presentation of the results is good. There is one point where the article could be improved - if the effect of DADS on cGMP levels could be demonstrated, if DADS is an inhibitor of PDE5 then this effect should be shown.
After minor revision it can be acceptable
Author Response
# Reviewer 4
The investigated phenomenon, namely the synergistic effect of two natural substance on apoptosis has of moderate interest, especially because it was investigated only on one cell type. However the following of the signalization route experimentally is valuable.
The experimental set-up is clear and the presentation of the results is good. There is one point where the article could be improved - if the effect of DADS on cGMP levels could be demonstrated, if DADS is an inhibitor of PDE5 then this effect should be shown.
The English of the article is problematic, with many sentences difficult to understand, confusing structure of sentences.
The logic of the introduction is somewhat lacking. In line 33, the claim that OHBFB would have anti-AML effects is not convincing based on only in vitro results. In the section after line 36, the correlation between the 67LR receptor and cGMP levels is missing. It is not clear why erectile dysfunction is highlighted in relation to PDE5 activity in a cancer related article.
In the methodology section, the description of the statistical parameters is missing, how many independent experiments were performed in each case.
For the results section:
- The term physiological concentration of OHBFB is not meaningful. Line 118 "DADS potentiated AML cell death caused by OHBFB, at an IC50 of 5.7 µM or 3.1 µM" is not understandable, presumably DADS concentrations are missing.
Response 1: In accordance with the Reviewer’s comment, we amended our manuscript as described below in Figure 1.
“We found that the IC50 of OHBFB alone was 10.8 mM, and the IC50 of DADS alone was 55.8 mM in HL60 cells (Figure 1A, B). Interestingly, the IC50 of OHBFB was 5.7 mM when treated with DADS at 10 mM (Figure 1C). Moreover, when treated with 20 mM DADS, the IC50 of OHBFB further decreased to 3.1 mM (Figure 1D). These results show that DADS potentiated the OHBFB-induced cell death effect in HL60 cells.”
- Slightly different parameters were used in the PBMCS cell viability analysis. A related question is what is the toxicity when using 25uM OHBFB? What is the reason for looking at viability at 96h for HL-60 cells and 72h for PBMCS cells?
Response 2: Thank you for your comment. Accordingly, we amended typo 96 h to 72 h in Figure legends.
- In figure caption, the SEM abbreviation is probably standard error of mean correctly cGMP-PDE enzyme activity - not meaningful term
Response 3: Thank you for your comment. Accordingly, we checked for SEM as described below in figure caption1.
“the standard error of the mean (SEM).”
- line 170 BAY 41-2272 - cell biological activity of this substance should be added in the text somewhere
Response 4: Thank you for your comment. Following your suggestion, we have made the additions as described below at result section 3.4.
" Bay 41-2272 is a soluble guanylate cyclase agonist, also recognized as an activator of intracellular cGMP production [3]."
- discussion refers to many things, could be a bit more concise. The exaggerated wording of the last sentences overstates the value of simple in vitro experiments. Nor is "physiological concentration" meaningful here.
Response 5: In accordance to the Reviewer’s comment, we removed “physiological concentration “and we amended our manuscript as described below at discussion section.
“Taken together, we conclude that DADS potentiates OHBFB-induced apoptotic cell death effect accompanied by the activation of the eNOS/cGMP/PKCδ/ASM sig-naling pathway in HL60 cells. As the effective experimental concentrations of combi-nation treatment these compounds are low concentrations than treat alone, the results are worthy of particular consideration. Moreover, the combination of OHBFB and DADS may be considered as promising candidates for the development of natural therapeutic drugs against AML.”

Round 2
Reviewer 1 Report
Comments and Suggestions for Authors
Paper is significantly improved. Still, a lot has been copied from authors previous publication. Please see title of the result 3.1 "Combination of EGCG and VDN Synergistically Induces Cell Death in Colorectal Adeno carcinoma" Its been taken from the previous publication (10.3390/cimb44120426) however the drug used in current MS is oolonghomobisflavan B (OHBFB) and diallyl disulfide (DADs). Its okay to do a parallel study as long as its not very similar and copy pasted like the results 3.1 title.The authors can cite their previous publication. I strongly recommend to work on the similarity.
Author Response
1. Paper is significantly improved. Still, a lot has been copied from authors previous publication. Please see title of the result 3.1 "Combination of EGCG and VDN Synergistically Induces Cell Death in Colorectal Adeno carcinoma" Its been taken from the previous publication (10.3390/cimb44120426) however the drug used in current MS is oolonghomobisflavan B (OHBFB) and diallyl disulfide (DADs). Its okay to do a parallel study as long as its not very similar and copy pasted like the results 3.1 title.The authors can cite their previous publication. I strongly recommend to work on the similarity.
Response : Thank you for your valuable suggestions. In response to the Reviewer's comments, we have revised the manuscript in the Result 3.1 title at line 131 and 146, as explained below, and addressed the similarity issue, reducing it to a 16% similarity index.
“Synergistic induction of cell death in HL60 cells by the combination of OHBFB and DADS”

Reviewer 2 Report
Comments and Suggestions for Authors
Some of my comments have been addressed. In advance to some point, the introduction and discussion still lack a crucial understanding. This can be elaborated on in the introduction and discussion. It is insisted on elaborate and cite the following study:
- Diallyl disulfide: A bioactive garlic compound with anticancer potential
Comments on the Quality of English LanguageMinor revisions
Author Response
- Some of my comments have been addressed. In advance to some point, the introduction and discussion still lack a crucial understanding. This can be elaborated on in the introduction and discussion. It is insisted on elaborate and cite the following study:
- Diallyl disulfide: A bioactive garlic compound with anticancer potential
Response : Thank you for your valuable suggestions. In accordance with the Reviewer’s comment, we have amended the manuscript at the introduction and discussion section as described below at line 43-49 and line 245-247.
“Abundant studies provide evidence that DADS, a bioactive garlic compound offers a range of health benefits, notably including its anticancer potential [15–18]. Previously, we identified H2S as a gasotransmitter that potentiates cancer cell death in multiple myeloma cells [17]. Intracellular H2S suppresses phosphodiesterase enzyme activity, thereby elevating cyclic nucleotide levels and promoting cancer cell death [13,17]. Here we showed garlic-derived a bioactive garlic compound DADS induced intracellular H2S production in HL60 cells.”
“Diallyl disulfide (DADS), a bioactive garlic compound recognized for its anticancer potential, is a primary organosulfur component derived from garlic (Allium sativum) and functions as a hydrogen sulfide (H2S) donor.”
